# An Association between Lower Extremity Function and Cognitive Frailty: A Sample Population from the KFACS Study

**DOI:** 10.3390/ijerph18031007

**Published:** 2021-01-23

**Authors:** Gwon-Min Kim, Bo-Kun Kim, Du-Ri Kim, Yung Liao, Jong-Hwan Park, Hyuntae Park

**Affiliations:** 1Health Convergence Medicine Laboratory, Biomedical Research Institute, Pusan National University Hospital, Busan 49241, Korea; rlarnjsals47@gmail.com (G.-M.K.); drkim4100@gmail.com (D.-R.K.); 2Faculty of Sports Health Care, In-je University, Gimhae 50834, Korea; fabulousbobo79@gmail.com; 3Department of Occupational Therapy, Graduate School of Inje University, Gimhae 50834, Korea; 4Department of Health Promotion and Health Education, National Taiwan Normal University, Taipei 106, Taiwan; liaoyung@ntnu.edu.tw; 5Department of Health Sciences, Graduate School, Dong-A University, Busan 49315, Korea

**Keywords:** cognitive frailty, Timed Up and Go test, Five Times Sit-to-Stand test, 6-min Walk Test, lower extremity function

## Abstract

The present study aimed to define the physical function and lipid profile for physical and cognitive frailty in community-dwelling Korean older adults. A total of 229 participants (age = 76.76 ± 3.72 years; body mass index = 24.83 ± 3.15) were classified into four groups: robust, pre-frailty, cognitive decline, and cognitive frailty. An analysis on the four groups was performed to measure their physical and cognitive function, as well as blood biomarkers. The area under (AUC) the receiver operating characteristic curve (ROC) indicated that the 6-min Walk Test (6MWT), Timed Up and Go test (TUG), and Five Times Sit-to-Stand test (FTSS) had the potential to distinguish the capacity of an old adult to predict cognitive frailty. The 6MWT had a higher sensitivity and the TUG and FTSS tests had greater specificity. With cognitive frailty as a categorical variable, cognitive frailty status was a significant predictor of the TUG (odds ratio (OR) 2.897; 95% confidence interval (CI), 1.283–6.541), FTSS (OR 3.337; 95% CI 1.451–7.673), and 6MWT (OR 0.204; 95% CI 0.070–0.591) tests. Our findings indicate that the 6MWT, TUG, and FTSS tests are closely related to cognitive frailty and can provide potential prognostic cutoff points.

## 1. Introduction

Frailty is associated with chronic diseases and poor prognoses, such as death or disability [1,2,3,4]. Among the multiple dimensions of frailty, physical frailty has been widely recognized as a public health priority in older adults, while the term “cognitive frailty” has been coined in the literature in recent years [5]. An international consensus group has proposed the concept of cognitive frailty as a heterogeneous cognitive state characterized by the simultaneous presence of both cognitive impairment and physical frailty, excluding concurrent or other dementias [6]. Recent research has shown that the relationship between cognitive decline and physical frailty has become more overt [7,8]. Furthermore, previous studies reported that cognitive frailty was related to a higher prevalence of obesity, low physical function, and disability rather than to physical frailty [9,10,11].

Regarding the relationship between physical and cognitive frailty, there is a reduction in physical function through cognitive decline, and this reduction in physical capacity reduces the functionality of old adults to perform complex motor tasks. This is related to a reduction in functional and impaired mobility [12]. Cognitive decline, in conjunction with declining physical function, has been related to an increased risk for dementia and disability in population-based research of older adults [13,14]. In particular, lower extremity weakness increases with age [15], resulting in substantial differences in lower extremity function between young and older adults [16,17,18]. Lower extremity weakness in older adults is also an independent predictor of future falls [19,20]. In addition, there is growing evidence that declined lower extremity function is also present in milder phases, which may be related to the level of cognitive decline [13,21].

However, a previous systematic review indicated that the findings of a relationship between changes in cognitive and physical functions are relatively small, owing to a few studies reporting an association [22]. In addition, it is unclear whether lower extremity weakness is associated with cognitive frailty in Korean older adults. Therefore, the present study aimed to compare the performance of different lower extremity function tests in well-characterized individuals with or without cognitive frailty, along with controls matched demographically and according to comorbidities, such as obesity and depression.

## 2. Materials and Methods

This study was approved by the Dong-A University Research Ethics Committee (IRB number is BR-009-019). All subjects provided their informed written consent prior to participation, which was obtained from all participants after they were provided with a detailed description of the experiment, in accordance with the ethical standards of the Declaration of Helsinki.

### 2.1. Participants

A total of 251 Korean older adults were recruited through an advertisement placed in the local health center and newspapers at Pusan, Korea. All applicants had to meet the following criteria to be included in the study: (1) age greater than 65 years; and (2) not diagnosed with a terminal illness. As shown in Figure 1, of the 251 initial participants, we excluded 19 based on the inclusion criteria. We then excluded three more applicants who were not eligible for the present study because they were illiterate, had a hip or knee joint arthroplasty within the preceding 6 months, or were unable to walk independently. In the end, 229 participants were included in the present study.

### 2.2. Cognitive Function

The Mini-Mental State Examination—Dementia Screening (MMSE-DS) [23] was used to screen for cognitive decline. When the score was <25, this was defined as cognitive decline. This test is one of the best known and most affordable cognitive screening instruments available for research and clinical practice [17].

### 2.3. Defined Cognitive Frailty

The five components of a physical frailty phenotype, which were assessed according to a pre-defined protocol, are self-reported exhaustion, unintentional weight loss, reduced grip strength, reduced gait speed, and self-reported low physical activity [18]. Depending on the number of criteria attained, individuals were classified as robust (0), pre-frail (1–2), and frail (3–5). However, in the present study, participants were divided into four groups as follows: (A) robust (physical frailty score of 0; MMSE score ˃ 25 points); (B) pre-frailty (physical frailty score ˃ 1; MMSE score ˂ 25 points); (C) cognitive decline (physical frailty score of 0; MMSE score < 25 points); and (D) cognitive frailty (physical frailty score > 1; MMSE score < 25 points) [9].

### 2.4. Body Composition

The body mass index and appendicular skeletal muscle mass index of all participants were measured using a bioelectrical impedance analysis. Body composition values were determined using an Inbody S10 device (S10; Inbody, Seoul, Korea, 2014). Waist circumference was measured at the narrowest part between the lower rib and the iliac crest (the natural waist) or, in case of an indeterminable waist narrowing, halfway between the lower rib and the iliac crest. On the other hand, hip circumference was measured over the widest part of the buttocks. Both measurements were recorded to the nearest half centimeter. The waist-to-hip ratio was calculated as the waist measurement divided by the hip measurement.

### 2.5. Blood Pressure and Blood Test

Arterial blood pressure was measured from the right arm in a seated position using a blood pressure monitor (HEM-7121; Omron Corporation, Kyoto, Japan, 2016). A blood analysis was performed on the plasma samples (5 mL) of the participants. A blood sample was drawn from an antecubital vein of participants between 08:00 and 10:00 a.m., after a 12–14 h overnight fast. The samples were collected in a seated position according to a standard protocol, and they were centrifuged within 30–45 min upon collection. Analysis of the samples was performed using auto-analyzers (Hitachi 902; Roche, Manheim, Germany, 1997). The samples were then collected into tubes containing clotting activators to isolate the serum. Triglyceride, total cholesterol, high-density lipoprotein cholesterol (HDL-C), low-density lipoprotein cholesterol, glucose, insulin, and creatinine kinase were analyzed using standard lipid profiles for older adults.

### 2.6. Physical Function

The hand grip strength of the dominant hand of participants was measured with a handheld dynamometer (TKK 5401; Takei Scientific Instruments, Tokyo, Japan, 2012). They were instructed to perform a maximal isometric contraction. During the assessment, the participants were asked to stand upright with their feet shoulder-width apart and to look forward with the elbow fully extended. The test was repeated after 30 s, and the highest value was recorded. The Five Times Sit-to-Stand test (FTSS) was then used to assess leg strength and endurance. Participants were instructed to cross their arms across the chest and stand up and sit down five times, as fast as possible, with the time measured in seconds [19]. For the Timed Up and Go test (TUG), related to leg function, a 3-m walkway was marked on a flat surface using a cone on the floor; the chair used had back support but no armrests (seat height: 46 cm). Participants were then asked to stand up from the chair, walk forward for the 3-m distance as rapidly as possible, walk back, and return to sit on the chair. Gait speed was measured in the standard gait task protocol, with the time measured in seconds. The gait speed test used a 7-m distance, which included a 1.5-m acceleration and deceleration distance; only the 4-m walk was timed for gait speed. Lastly, the 6-min Walk Test (6MWT) was used to examine the aerobic capacity of the participants since the test is specifically tailored and self-paced for older adults [20]. Each participant was instructed to walk as far as possible accompanied, to avoid any interference in walking speed, along a 20 m flat, obstruction-free corridor with one chair on both ends of this corridor. Participants were then instructed to walk around the cones at a fast speed for 6 min; not to run or jog. The hand grip strength and gait speed are some of the essential variables to screen for frailty [18]. The validity and reliability of all of the tests that are employed in the present study were adequately proved via lots of previous studies that investigated the physical function of older adults [24,25,26].

### 2.7. Statistical Analysis

The statistical analysis was carried out using the IBM SPSS Statistics for Windows version 21.0. (IBM Corp: Armonk, NY, USA). The characteristics of the study population are presented as the mean ± standard deviation (SD) for continuous variables and as proportions for categorical variables. One-way analysis of variance (ANOVA) was carried out to compare the anthropometric measures, physical function, and biomarker differences between the four groups. Post-hoc Bonferroni tests were performed after one-way ANOVA to look for differences between the groups, physical functions, and blood biomarkers. Subsequently, we used logistic regression analyses to determine the odds ratio and 95% confidence intervals, adjusted for sex and age, and to assess the independent associations between physical function and the risk of cognitive frailty. Statistical significance was defined as *p* < 0.05. The best cutoffs for predicting physical function to cognitive frailty were evaluated by the parameters (6MWT, TUG, and FTSS) provided by the receiver operating characteristic curve (ROC), the area under the ROC curve (AUC), sensitivity, and specificity. Analyses were performed using MedCalc for Windows version 9.1.0.1 (MedCalc^®^Software Corporation, Mariakerke Ostend, Belgium, http://www.medcalc.be).

## 3. Results

### 3.1. Baseline Characteristics of the Participants

The mean (SD) age of the participants was 76.7 (3.7) years, and 80.3% of participants were women. Table 1 shows the measures (mean and SD) for the physical, cognitive, functional, and blood biomarkers in the present study.

### 3.2. Comparisons of Physical Function and Blood Biomarkers Using ANOVA Followed by Pairwise Comparisons Using the Bonferroni Test.

As a result of comparing the physical function of the participants in terms of the four groups, a significant difference was found in the 6MWT (*p* < 0.02), FTSS (*p* < 0.00), and TUG (*p* < 0.00) levels. A post-hoc test showed that the values of the 6MWT were significantly lower in group A4 (cognitive frailty) than in group A1 (robust); the FFSS values were significantly lower in group A3 (cognitive decline) and A4 compared to group A1; and that the TUG values were significantly lower in group A3 and A4 compared to group A1, while that of group A4 was significantly lower compared to group A2 (Table 2).

### 3.3. For Multi-Nominal Logistic Regression Models Predicting Cognitive Frailty from Physical Function

The results of the logistic regression analyses are shown in Table 3. With cognitive frailty as a categorical variable, TUG status was a significant predictor at ≥8.21 (OR 2.790, 95% CI 1.266–6.151; adjusted model: OR 2.897, 95% CI 1.283–6.541); FTSS status was a significant predictor at ≥12.54 (OR 3.115, 95% CI 1.390–6.982; adjusted model: OR 3.337, 95% CI 1.451–7.673); and 6MWT status was a significant predictor at ≥420.01 (OR 0.241, 95% CI 0.088–0.662; adjusted model: OR 0.204; 95% CI, 0.070–0.591).

### 3.4. ROC Curves of 6MWT, TUG Test, and FTSS Test in the Cognitive Frailty Groups

We observed that an optimal relationship between sensitivity and specificity was achieved at a 6MWT of <392.1, TUG > 8.1, and FTSS > 12.47. For this cut-off point, the ROC analysis revealed a sensitivity of 83.61% and specificity of 47.62%, a sensitivity of 44.26% and specificity of 79.76%, and a sensitivity of 45.90% and specificity of 80.95% for 6MWT, TUG, and FTSS, respectively (Figure 2).

## 4. Discussion

In the present study, cognitive frailty risk was significantly associated with the 6MWT, TUG, and FTSS scores. The participants with cognitive frailty had a lower physical function than those with cognitive decline or physical frailty; however, the association between cognitive frailty and lipid parameters was not observed in this study sample. This finding is inconsistent with the results of previous studies, which demonstrate that, in addition to inflammatory status, some lipid parameters could also be convincingly related to cognitive decline and frailty syndrome [21,22]. With this being said, cognitive frailty prediction is perceived as difficult to identify from simple blood tests. These discrepant results were obtained owing to the differences in frailty assessment criteria used to determine frailty status.

Our study supports the new definition of physical function in older adults with cognitive frailty. The AUC (Figure 2) indicated that the 6MWT, TUG, and FTSS tests had the potential to determine the capacity of an older adult in order to predict cognitive frailty; the resulting cutoff points were ≤392.1 for the 6MWT, >8.1 for the TUG test, and >12.47 for the FTSS test. The 6MWT had a higher sensitivity, while the TUG and FTSS tests had greater specificity. Prior studies on cutoff points and poor performance on the 6MWT ( ≤ 400 m), TUG test ( ≥ 9 s), and FTSS test ( ≥ 10 s) were useful for stratifying the risk of disability development in community-dwelling older adults [25,27]. Although the previously used cutoff point was different from the one determined by the 6MWT, the difference was not very large [27]. In addition, the FTSS test had a higher cutoff point than the prior study [25]. This result may be due to the fact that the study had more female participants since it is known that women have lesser muscle performance than men [28]. Moreover, this study had a lower cutoff point for TUG than the previous study. The present study had a specificity of 79.56% for the cutoff point of 8.1 for the TUG test, which is only slightly better than the specificity of the previous study [25]. Additionally, the differences in ethnicity might play a role in the differences in the measurement of the resulting cutoff value [29,30]. Therefore, this finding suggests using a cutoff point for cognitive frailty specific to older Korean adults.

We investigated whether the new cutoff point identified in the present study was associated with cognitive frailty in older adults in the community. The results showed significant associations between the TUG test (OR 2.897, 95% CI 1.283–6.541), FTSS test, (OR 3.337, 95% CI 1.451–7.673), and 6MWT (OR 0.204, 95% CI 0.070–0.591) for the cognitive frailty group (Table 3). This result was consistent with previous studies showing that the cognitive frailty group had a negative impact on the level of physical function in older adults, which was better than a robust group. Physical frailty includes low physical function, and one’s initial cognitive and physical health have been shown to be associated with their subsequent cognitive and physical decline [31]. Therefore, the results showed a higher ratio owing to synergies in physical frailty and cognitive decline. In the present study, an intimate relationship existed between physical function (i.e., low leg strength and low aerobic capacity) and cognitive frailty.

Finally, previous studies suggested that different cognitive frailty groups may be related to an increased risk of functional disability, worsened quality of life, hospitalization, and all-cause mortality. Some large longitudinal population-based studies have shown that different cognitive frailty models may increase the risk of dementia, vascular dementia, and neurocognitive disorders [32]. For different groups of cognitive frailty, physical frailty may precede the onset of cognitive impairment [33]; thus, the present study is useful for older adults for detecting physical frailty.

However, the present study has some limitations. First, a cross-sectional design was used; therefore, the findings do not illustrate any causal relationships between the examined variables. Second, forming generalizations based on this study is difficult as more women were recruited compared to men. Future research should thus consider the ratio of men and women. Third, this study had more individuals under pre-frailty than frailty. Therefore, it must be considered that if there were more participants in the frailty group, the results might have been different.

## 5. Conclusions

In conclusion, our findings indicate that the 6MWT, TUG test, and FTSS test are closely related to cognitive frailty and may be used to determine potential prognostic cutoff points. However, further studies are needed to better define the related cognitive frailty in a multicenter study with a larger sample size. With more research, preventative and new strategies may be developed.

## Figures and Tables

**Figure 1 ijerph-18-01007-f001:**
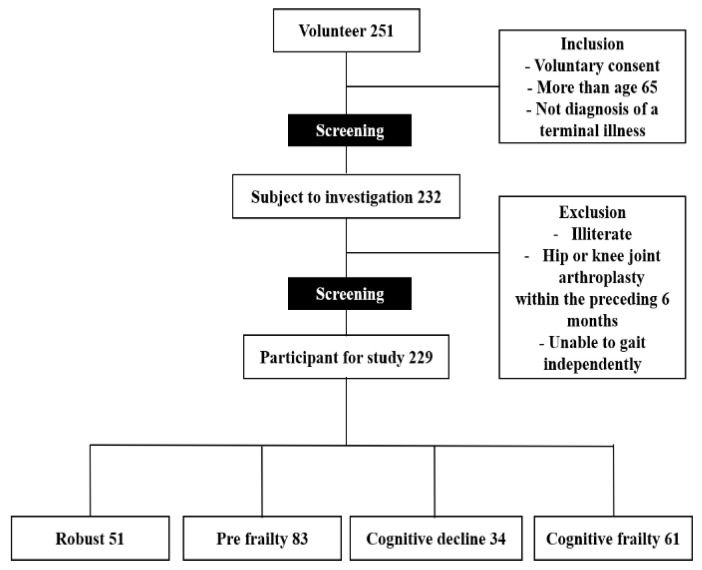
Study flow diagram.

**Figure 2 ijerph-18-01007-f002:**
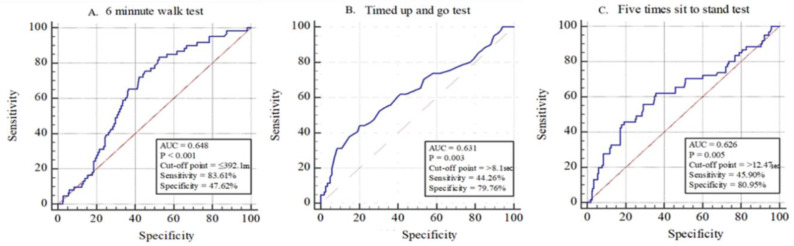
Receiver operating characteristic (ROC) curves of the 6MWT, TUG test, and FTST test in the cognitive frailty groups in relation to the four groups (Blue line: Physical function area; Red line: Reference).

**Table 1 ijerph-18-01007-t001:** Baseline characteristics of the participants (n = 229).

Basic Factor	
Female (no, %)	184 (80.3)
Age (year)	76.76 ± 3.72
Body mass index (kg/m^2^)	24.83 ± 3.15
Appendicular skeletal muscle (kg)	15.40 ± 3.15
Waist-hip ratio (%)	0.93 ± 0.07
Systolic blood pressure (mmHg)	138.81 ± 18.00
Diastolic blood pressure (mmHg)	74.18 ± 11.22
Cognitive function	
Mini mental state examination (point)	24.74 ± 3.54
Physical function	
6 minute walk test (m)	378.11 ± 63.25
Five times sit to stand test (s)	10.96 ± 3.60
Timed up and go test (s)	7.33 ± 1.52
Grip strength (kg)	22.03 ± 5.74
Gait speed (m/s)	1.14 ± 0.18
Bio-marker	
Total cholesterol (mmol/L)	4.76 ± 4.76
Triglycerides (mmol/L)	1.45 ± 0.64
Low density lipoprotein cholesterol (mmol/L)	3.07 ± 0.91
High density lipoprotein cholesterol (mmol/L)	1.40 ± 0.35
Glucose (mmol/L)	2.55 ± 0.46
Insulin (mmol/L)	8.41 ± 6.01
Creatinine kinase (μkat/L)	1.74 ± 2.24
Data represent mean ± SD and number, %	

**Table 2 ijerph-18-01007-t002:** Comparison of physical function and blood biomarkers in old adults for 4 groups using a one way ANOVA followed by pair-ways comparison using a Bonferroni correction.

Physical Function	A 1 (51)	A 2 (83)	A 3 (34)	A 4 (61)	*p*	Post-Hoc
6 min walk test (m)	397.27 ± 65.11	381.77 ± 74.94	372.65 ± 47.61	360.16 ± 45.52	0.02	A 1 > A 4
Five times sit to stand test(s)	9.69 ± 2.41	10.60 ± 3.63	11.84 ± 3.79	12.01 ± 3.92	0.00	A 1 < A 3, A 4
Timed up and go test (s)	6.74 ± 1.10	7.10 ± 1.37	7.74 ± 1.45	7.92 ± 1.80	0.00	A 1 < A 3, A 4; A 2 < A 4
Grip strength (kg)	22.70 ± 4.59	22.68 ± 6.03	20.97 ± 4.96	21.18 ± 6.49	0.24	-
Gait speed (m/s)	0.87 ± 0.12	0.92 ± 0.19	0.89 ± 0.11	0.91 ± 0.13	0.21	-
**Biomarker**						
Total cholesterol (mmol/L)	4.77 ± 0.99	4.75 ± 0.94	4.93 ± 1.00	4.68 ± 0.86	0.63	-
Triglycerides (mmol/L)	1.37 ± 0.59	1.48 ± 63.90	1.54 ± 0.62	1.43 ± 0.56	0.67	-
Low density lipoprotein cholesterol (mmol/L)	3.07 ± 0.95	3.05 ± 0.93	3.23 ± 0.98	3.01 ± 0.83	0.75	-
High density lipoprotein cholesterol (mmol/L)	1.44 ± 0.33	1.39 ± 0.36	1.37 ± 0.38	1.38 ± 0.36	0.77	-
Glucose (mmol/L)	5.33 ± 0.90	5.62 ± 1.10	5.60 ± 0.86	5.32 ± 0.92	0.18	-
Insulin (mmol/L)	8.10 ± 5.03	7.61 ± 5.47	10.56 ± 8.02	8.56 ± 6.04	0.11	-
Creatinine kinase (μkat/L)	2.23 ± 3.11	1.84 ± 2.67	1.55 ± 0.96	1.31 ± 0.62	0.16	-

A 1= Robust; A 2 = pre frail; A 3 = Cognitive decline; A 4 = Cognitive frail; One-way analysis of variance was used with a post hoc Bonferroni correction for multiple comparisons; Data represent mean ±SD. *p* valued less than 0.05.

**Table 3 ijerph-18-01007-t003:** Four multi-nominal logistic regression models predicting physical function in cognitive frailty.

	Robust Model	Adjust Model
Timed up and go	odds ratio (CI 95%)	odds ratio (CI 95%)
< 6.31	Ref.	Ref.
6.31–7.10	0.748 (0.295–1.897)	0.772 (0.302–1.972)
7.11–8.20	1.080 (0.458–2.546)	1.120 (0.464–2.703)
≥ 8.21	2.790 (1.266–6.151)	2.897 (1.283–6.541)
FTSS	odds ratio (CI 95%)	odds ratio (CI 95%)
< 8.69	Ref.	Ref.
8.69–10.37	0.736 (0.293–1.848)	0.743 (0.296–1.870)
10.38–12.53	0.828 (0.336–2.040)	0.856 (0.345–2.122)
≥ 12.54	3.115 (1.390–6.982)	3.337 (1.451–7.673)
6MWT	odds ratio (CI 95%)	odds ratio (CI 95%)
< 340.01	Ref.	Ref.
340.01–377.80	1.604 (0.752–3.422)	1.587 (0.739–3.410)
377.81–420.00	0.491 (0.208–1.156)	0.450 (0.187–1.080)
≥ 420.01	0.241 (0.088–0.662)	0.204 (0.070–0.591)

FTSS: Five timed sit to stand; 6MWT: 6-minnute walk test; CI: confidence interval; Adjust model: sex and age.

## Data Availability

Qualified researchers can obtain the data from the corresponding author (htpark@dau.ac.kr). The data are not publicly available due to privacy concerns imposed by the IRB.

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
