# Peer review of "An Association between Lower Extremity Function and Cognitive Frailty: A Sample Population from the KFACS Study"

_ijerph, 2021, doi:10.3390/ijerph18031007_

Round 1

Reviewer 1 Report

General

The present study reports an association between lowered physical function and categories of cognitive decline. However, the way it is written, in places, it seems that cognitive decline predicts a physical decline. This needs clarifying in the paper. The manuscript needs much more careful writing so that the methods section is completely clear, namely how the different categories (e.g. robust, pre-frailty, cognitive decline, cognitive frailty) were defined and how they may update / differ from earlier definitions of cognitive frailty. Details about how recruitment to the study proceeded needs further clarification (e.g. recruitment via local newspaper, radio station, social media and so on). Further detail needs to be provided over the 6 min walk test, and when and how blood pressure was measured. Also how the blood was treated prior to analysis. A rationale for the chosen blood biomarkers should be provided. More background should be provided on the receiver operating characteristic (ROC) and how sensitivity and specificity factors were identified from this analysis. What were the main findings of the blood analysis? Does that add anything to the present findings? (If not this can also be stated). Admittedly English is not the first language, having said that, the manuscript is poorly written and needs proof reading by a native English speaker prior to submission. The authors have not been consistent throughout e.g. TUG and tug can be found in the text.

Here are some suggestions to improve clarity of the written text.

Specific

  1. Title: An association between lower extremity function and cognitive frailty: a sample population from the KFACS study.
  2. Abstract: Background: The aim of present study was to assess function and lipid profile of physical and cognitive frailty in community dwelling Korean elders. Methods: 229 old adults (insert age, height, body mass mean and SD)…..

19….for measures of physical and cognitive function, together with blood biomarkers was investigated.

  1. Conclusion: our findings indicate that 6MWT, TUG, FTSS are closely associated with cognitive frailty and it (??) – what is it?
  2. Introduction:
  3. ….in the older adults and the term ‘cognitive frailty’ has been coined in the literature in recent year.

39….cognitive frailty was more strongly (?) related to a higher prevalence of obesity…….than physical frailty per se [9-11].

  1. Regarding the relationship between physical and cognitive frailty, the reduction of physical function and cognition declines as physical capacity reduces the functionality of old adults to perform complex motor tasks and this was related to a reduction in function and impaired mobility [12]. This sentence is too complex and too long. It needs separating into two shorter, simpler sentences.
  2. …extremity weakness increases with age

47…between young and old adults

48-49. There is growing evidence that lower extremity disturbances …(what are lower extremity disturbances? Please provide an example).

52… physical function

52-53.  reported an association [22].

  1. Also, until recently no defined had been had been located on the lower extremity weakness that rerated between cognitive frailty in Korean old adults. (Needs re-writing for clarity).

55-57…of individuals with or without cognitive frailty, together with demographically matched controls, matched for co-morbidities such as obesity and depression.

  1. Materials and Methods

59-60. Insert a sentence to state that subjects were treated in accordance with the principles laid down in the Declaration of Helsinki (1986).

2.1 Participants

How were participants recruited? Please insert method(s) of recruitment.

63….to the inclusion and exclusion criteria

  1. …voluntary consent (2) more than 65 years old (3) not diagnosed with a terminal illness (these changes also need to be changed in Figure 1).

65-66…. unable to walk independently

66-69. The five components of a physical frailty phenotype comprise: self-reported exhaustion……….that were assessed according to a pre-defined protocol [17].

69-70. …the number of criteria attained individuals were classified as…

70-71 Mini Mental State Examination (MMSE) was tested using determined by Global cognitive. (Needs re-writing for clarity).

72-74 In the present study cognitive frailty was defined as ….(define clearly)

75-76 Also, physical frail, with no cognitive decline was defined as having pre frailty alone [9]. (Needs re-writing for clarity).

2.2.1. Physical function

  1. Hand grip strength was measured with a handheld dynamometer (TKK 5401, Tokyo, Japan) using the dominant hand.

How was appendicular skeletal muscle mass measured?

Which instrument was used to measure blood pressure? How many minutes was allowed for blood pressure to stabilize before readings were taken? Was blood pressure measured before a blood draw? Where was the blood draw taken from?

How long was the corridor for measuring distance covered during the 6 minute walk test? Were frail participants accompanied in case of a fall? What safety measures were in place?

83-86 The five times sit to stand test (FTSS) was used to assess leg strength / endurance. Participants were instructed to cross their arms across the chest and stand up and sit down 5 times, as fast as possible, with time measure in seconds.

87-88. A 3-m walkway was marked on a flat surface for the timed up and go test (TUG), the chair used had a back-support with no arm rests (seat height, 46 cm).

2.2.2. Blood test

  1. A blood sample was drawn (anatomical location?) from participants between 08:00-10:00 after a 12-14 hour overnight fast.

What was the volume of blood volunteered by participants?

What was the rationale for choosing these particular blood bio-markers?

State if the blood analysis was performed on plasma samples (how much, ml) for analysis.

105-106…..look for differences between groups, physical function and blood bio-markers.

110 P<0.05

112 …sensitivity and specificity need defining in the context of the ROC analysis. What parameter defines sensitivity, what parameter defines specificity?

  1. Results

3.1

  1. The mean (SD) age of participants was 76.7 (3.7) years with 80.3% being women. Table 1 shows the measures (mean and SD) for the physical, cognitive, functional and blood bio-markers in the present study. (Text 117-121 can be removed – as this duplicates information in Table 1).
  2. 3.2 Comparisons of physical function and blood bio-markers…….

125-128. Sentence beginning ….While ANOVA also negative an increase in 6MWT…..(Table 2). (Needs re-writing for clarity).

  1. 3.3 Four multi-nominal logistic regression models predicting cognitive frailty from physical function.
  2. Discussion

148      In the present study…

157-158 We found that a considerable new definition of physical function is present among old adults with cognitive frailty. (Needs re-writing for clarity).

163-165 Our 6MWT cutoff point is different from the one proposed by the prior study but, there was not much different from the prior study. However, this study than a prior study spears the high cutoff point for FTSS. (Needs re-writing for clarity).

168-169 This study was cutoff point of 8.1 and a specificity of 79.56 for a tug. The cutoff point was the value of lower and the specificity was percent of higher than the result of the previous study [24]. (Needs re-writing for clarity).

173 We investigated whether the new cutoff point identified in the present study…

186-188 Physical frailty may precede the onset of cognitive impairment [31], thus the present study is meaningful for old adults as a means of detecting physical frailty.

189 The present study has some limitations.

190-191 More women were recruited to the study compared with men.

  1. Conclusions

196-197 Especially, there was a high correlation between the ratios 6mwt among physical function in cognitive frailty.

This is the first time correlation has been mentioned – it should appear earlier (results / discussion sections).

References

  1. Morley, J. E. et al. – all authors need to be acknowledged in this section (Refs 1-31)

215 Journal titles e.g. Age and Ageing – need capitals throughout (Refs 1-31).

Table 1

Female (no.; %)

Insert data for mem (number and %)

Blood bio-markers – please use units mmol/l

  1. 3.2 Comparison of physical function and blood biomarkers in old adults for 4 groups using a one way ANOVA followed by pair-ways comparison using a Bonferroni correction.

Table 2 Title needs amending for clarity.

Q1, Q2, Q3, and Q4 may be confused with quartiles – suggest using A1, A2, A3, and A4.

Table 3        Four multi-nominal logistic regression models predicting physical function in cognitive frailty.

Figure 2 These figures are far too small and need to be much bigger for clarity.

Author Response

Thank you for comment.

Reviewer 2 Report

rewrite requires to raise standard of English presentation.

The study is of interest, however requires a careful edit. 

I think the conclusions, could go further to explain and understand the link with the outcome measures and frailty, or more simply cognitive decline quickly becomes physical decline or frailty. 

Author Response

Thank you for comment.

Reviewer 3 Report

  1. THERE ARE ERRORS IN THE SYNTAX, THUS LOOSING THE MEANING i.e. page 1, line 42; page 2, line 42; page 2, line 52; line 73,74; page 6, line 152, line 154, line 198.
  2. by definition for cognitive decline group you set a cutoff for MMSE score >24. What was the upper limit? for exable MMSE =28 could be considered as cognitive decline? Please explain and be more accurate.
  3. the whole paper needs language refinement

Author Response

Thank you for comment.

Reviewer 4 Report

Thanks for allowing me to review the manuscript which aimed to defined physical function and lipid profile for physical 17 frailty and cognitive frail in community-dwelling Korean old adults

Abstract and Keyword: Correct.

Introduction. It is ok, but the introduction should focus more on the relationship between changes in cognitive and physical functions. The studies found should be reported. The main point of the introduction is correct however systematic review points outs should be specified.
Methods.
Correctly. MMSE should be specified if it has been adapted to the population of Korea.
Also "Time up and go test" should be explained.
Front another point of view why have not been evaluated the patients with the Tinetti test?? Different studies show us more specificity in the Tinetti test.

Discussion.
Should be improved according in the text there is not enough information in blood tests and frailty. Please, try to identify more studies about the blood test.
Besides explaining more the optimal relationship in the discussion at 6MWT of ≤3 141 92.1 and TUG of > 8.1, FTSS of > 1 2.47. try to identify studies who can approach it as an affirmation

Author Response

Thank you for comment.

Round 2

Reviewer 1 Report

Thank you for making the methods section much clearer. There are still some points in the text that need changing - to enhance clarity for the reader. I attach a word document where I make suggestions to improve clarity.

Reviewer 2 Report

I have considered the edit and revised manuscript. i can only view the manuscript with the track changes present, in  a pdf.

I remain concerned about the presentation and consideration of methodological detail
